# A dedicated visual pathway for prey detection in larval zebrafish

Julia L Semmelhack[1], Joseph C Donovan[1,2], Tod R Thiele[1], Enrico Kuehn[1], Eva Laurell[1], Herwig Baier[1]*

[1]Department Genes-Circuits-Behavior, Max Planck Institute of Neurobiology, Martinsried, Germany; [2]Program in Neuroscience, University of California at San Francisco, San Francisco

**Abstract** Zebrafish larvae show characteristic prey capture behavior in response to small moving objects. The neural mechanism used to recognize objects as prey remains largely unknown. We devised a machine learning behavior classification system to quantify hunting kinematics in semi-restrained animals exposed to a range of virtual stimuli. Two-photon calcium imaging revealed a small visual area, AF7, that was activated specifically by the optimal prey stimulus. This pretectal region is innervated by two types of retinal ganglion cells, which also send collaterals to the optic tectum. Laser ablation of AF7 markedly reduced prey capture behavior. We identified neurons with arbors in AF7 and found that they projected to multiple sensory and premotor areas: the optic tectum, the nucleus of the medial longitudinal fasciculus (nMLF) and the hindbrain. These findings indicate that computations in the retina give rise to a visual stream which transforms sensory information into a directed prey capture response.

## Introduction

The visual systems of many species have an innate capacity to respond to features that denote either prey or predators (*Olberg et al., 2000*; *Ewert et al., 2001*; *Simmons et al., 2010*; *Yilmaz and Meister, 2013*). However, the circuits underlying these responses are mostly unknown. Zebrafish larvae have an instinctive ability to hunt small moving prey objects, such as paramecia, as soon as they start to swim at five days post fertilization (5 dpf). Before initiating a prey capture swim, a larva must select the target from its surroundings, calculate its location, and make a decision as to whether the target is worth pursuing. It then initiates a multi-step motor routine involving bouts of turning and swimming toward the prey, culminating in a consummatory strike (*Budick and O'Malley, 2000*; *Borla et al., 2002*; *Gahtan et al., 2005*; *McElligott and O'Malley, 2005*). Precise maneuvers are required, and so prey capture tail movements are quite different from those observed during routine swims or escapes. To orient towards a paramecium on the left or right, larvae perform j-turns—unilateral bends where the tail is held in a J shape. If the prey is directly ahead, they slowly swim toward it, with back and forth undulations of the tail (*Patterson et al., 2013*). These movements appear to be triggered by small moving objects, but it is unclear how or where in the brain these objects are identified as prey.

Under the appropriate conditions, larvae can be induced to perform hunting swims when they are partially restrained in agarose. A head-fixed preparation facilitates automated tracking of tail movements, stimulus control, and functional imaging. Two recent studies have shown that a moving dot can evoke eye convergence and approach swims in head-fixed larvae (*Bianco et al., 2011*; *Trivedi and Bollmann, 2013*). However, these studies did not investigate the specific tail kinematics evoked by prey stimuli, or seek to categorize the tail movements as distinct from other types of swims. Here, we identified the distinctive features of prey capture swims in head fixed larvae, and developed a machine learning algorithm to quantify and distinguish these swims from other types of behavior. This allowed

*For correspondence: hbaier@neuro.mpg.de

Competing interests: The authors declare that no competing interests exist.

**eLife digest** Our ability to recognize objects, and to respond instinctively to them, is something that is not fully understood. For example, seeing your favorite dessert could trigger an irresistible urge to eat it. Yet precisely how the image of the dessert could trigger an inner desire to indulge is a question that has so far eluded scientists. This compelling question also applies to the animal kingdom. Predators often demonstrate a typical hunting behavior upon seeing their prey from a distance. But just how the image of the prey triggers this hunting behavior is not known.

Semmelhack et al. have now investigated this question by looking at the hunting behavior of zebrafish larvae. The larvae's prey is a tiny microbe that resembles a small moving dot. When the larvae encounter something that looks like their prey, they demonstrate a hardwired hunting response towards it. The hunting behavior consists of a series of swimming maneuvers to help the larvae successfully capture their prey.

Semmelhack et al. used prey decoys to lure the zebrafish larvae, and video recordings to monitor the larvae's response. During the recordings, the larvae were embedded in a bed of jelly with only their tails free to move. The larvae's tail movements were recorded, and because the larvae are completely transparent, their brain activity could be visually monitored at the same time using calcium dyes.

Using this approach, Semmelhack et al. identified a specific area of the brain that is responsible for triggering the larvae's hunting behavior. It turns out that this brain region forms a circuit that directly connects the retina at the back of the eye to nerve centers that control hunting maneuvers. So when the larva sees its prey, this circuit could directly trigger the larva's hunting behavior. When the circuit was specifically destroyed with a laser, this instinctive hunting response was impaired.

These findings suggest that predators have a distinct brain circuit that hardwires their hunting response to images of their prey. Future studies would involve understanding precisely how this circuit coordinates the larvae's complex hunting behavior.

us to quantify the tail movements produced in response to a range of artificial prey stimuli, and determine the ideal stimulus to evoke the prey capture response.

To begin to identify the neural circuits for prey identification, we focused on the retinal ganglion cells (RGCs), the output neurons of the retina. There is a precedent for RGCs acting as prey detectors—a classic study in the frog found RGCs that responded to small (1–3°) objects moving through the visual field (*Lettvin et al., 1959*). In the mammalian visual system, certain RGC types act as feature detectors for one aspect of the visual scene, and the axons of these RGCs project to visual nuclei that mediate the response to that feature. For example, intrinsically photosensitive RGCs project to nuclei that control circadian rhythms and the pupillary light reflex (*Chen et al., 2011*), and RGCs that respond to whole field motion innervate nuclei that drive compensatory eye movements (*Simpson, 1984*; *Dhande et al., 2013*). In zebrafish, RGC axons terminate in ten retinorecipient areas called arborization fields (AFs) (*Burrill and Easter, 1994*), most of which have not been functionally characterized. We hypothesized that there could be a class of RGCs that respond specifically to prey-like visual stimuli and project to one or more of these AFs.

By imaging RGC axons in the AFs, we identified one visual area, AF7, which responds specifically to the optimal artificial prey stimulus, as well as to actual paramecia. Targeted laser ablation showed that this area is important for the behavior. Finally, we found that neurons with arbors in the AF7 neuropil innervate multiple areas known to be involved in prey capture behavior and locomotion in general—the optic tectum, nMLF, and hindbrain. These results identify AF7 as vital part of the prey capture pathway, and link a dedicated retinal output to an ecologically relevant behavior.

## Results

### Analysis and classification of prey capture swims

To investigate how prey objects are identified, we developed a head-fixed prey capture assay and automated behavioral classification system. Larvae were embedded in agarose at 6 dpf, and their tails were freed so that swimming movements could be recorded with a high-speed camera. Visual

stimulation was provided by a small OLED screen in front of the larva. In preliminary experiments, we found that a small (~3° of the visual field) white dot moving horizontally across a black screen was effective in evoking behavior. Larvae performed two types of behavior in response to this virtual prey: forward swims, which consisted of low amplitude oscillations of the tail, and j-turns (*Figure 1A*) (*McElligott and O'Malley, 2005*). These swims were accompanied by eye convergence, which is another kinematic feature of prey capture (*Bianco et al., 2011*; *Patterson et al., 2013*). Larvae also performed spontaneous swims in the absence of visual stimuli. To analyze tail movements, we digitized the tail, using a custom algorithm to assign ~40 points along its length in each frame (*Figure 1B*). Plotting the position of the tip of the tail during prey capture and spontaneous swims over time revealed several kinematic differences between the behaviors (*Figure 1C*). Prey capture forward swims and spontaneous swims both consist of back and forth movements of the tail, but the amplitude of the prey capture swim is much lower. In contrast, during a j-turn the tail is deflected to one side, often for hundreds of milliseconds.

The apparent differences between spontaneous swims and prey capture were confirmed when we analyzed several hundred expert-classified prey capture and spontaneous swim videos. The average amplitude of the tail movements for prey capture bouts was 17% of tail length, while for spontaneous swims it was 48% of tail length (*Figure 1D*). We also observed the sustained tail deflections characteristic of j-turns. Plotting the duration of the longest tail deflection for each bout reveals a single peak at 27 ms for the spontaneous swims (*Figure 1E*). For prey capture swims, we also see a peak at this duration, representing the back and forth motion during forward swims, but in addition we see a long shoulder of turns of much longer duration, which consists of j-turns (*Figure 1E*). These stark differences suggest that prey detection triggers a specialized motor program.

In order to characterize the stimuli that evoke prey capture, we needed an objective method to quantify the behavior and distinguish it from spontaneous swims. We used a support vector machine (SVM), a supervised learning algorithm that allows for multi-dimensional classification, to train a classifier, which could then categorize new data. For this classification, prey capture forward swims and j-turns were not distinguished; both were included within the category of prey capture.

We trained the SVM on a set of 369 expert-labeled and digitized videos (*Figure 1F*). Bouts of swimming within the videos were identified using a thresholding operation on the smoothed derivative of the tail bend angle. For the dimensions of the SVM, we choose relatively simple parameters that allowed us to collapse a feature of the bout (e.g. tail curvature) into a single number. During training, each bout was assigned a position in 5-dimensional space corresponding to its values for the 5 parameters (*Figure 1—figure supplement 1*), and decision boundaries were drawn in multidimensional space to separate the two types of behavior. The trained SVM could then be used to classify new videos. Using this approach, we were able to achieve a cross-validated accuracy of 96% (*Figure 1—figure supplement 1*).

## Identifying the ideal prey stimulus

We next used the virtual prey assay and the SVM classifier to determine the ideal stimulus for evoking prey capture behavior. We presented head-fixed larvae with white dots ranging in size from 0.5–30° in diameter, while recording tail movements. To quantify the behavior, we used the trained SVM to identify swim bouts as prey capture or spontaneous swims. We calculated a prey capture score based on the percentage of time during each trial that the larva performed prey capture bouts. We found that a 3° dot was the optimal size to trigger prey capture (*Video 1*), and that the behavior was strikingly reduced when we increased the size to 10° (*Figure 2A*), which is consistent with data from free swimming and head-fixed larvae (*Bianco et al., 2011*; *Trivedi and Bollmann, 2013*). We also tested the response to dots traveling at a wide range of speeds, and found that 90°/s is the ideal speed for prey stimuli, and that larvae respond minimally to stimuli moving slower than 12°/s or faster than 360°/s (*Figure 2B*).

## Functional imaging of RGC axons

Given the highly selective nature of this innate behavioral response, we hypothesized that the preference for prey stimuli of a certain size and speed might arise in the retina. In that case, there should be a population of RGCs that responds specifically to prey. A recent functional imaging study reported responses to prey in the largest AF, the optic tectum (*Muto et al., 2013*). However, the tectum receives input from many different types of RGCs, and responds to a wide variety of stimuli other than prey

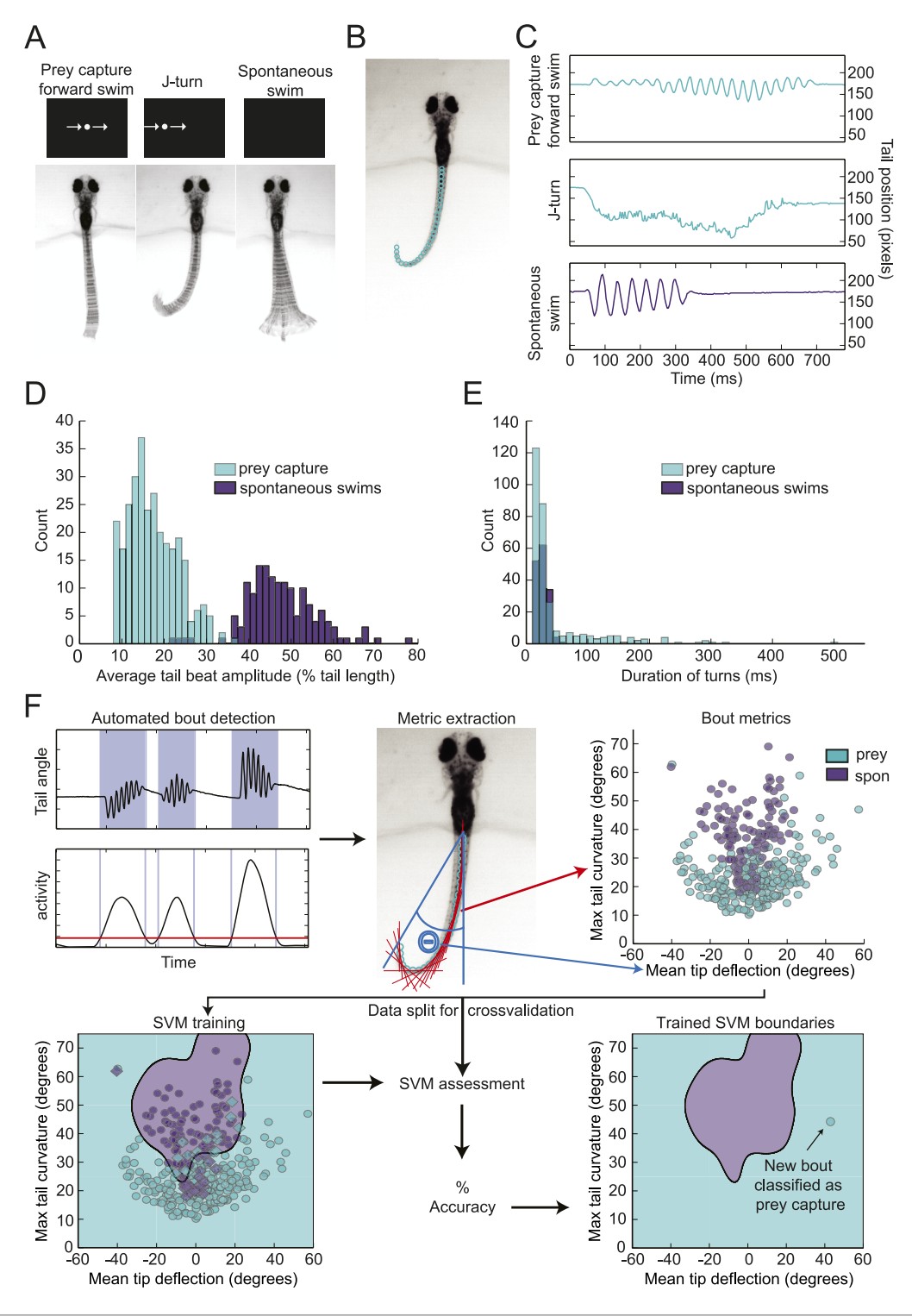

**Figure 1**. Head-fixed larvae respond to virtual prey with distinctive swimming movements. (**A**) Overlay of 50 frames (167 ms) of high-speed video showing examples of behavior in head fixed larvae. Larvae performed forward swims in response to a 3° dot. j-turns were observed when the same dot was to the right or left. Spontaneous swims were often observed in the absence of any stimulus. (**B**) Example video frame showing points assigned by the digitization algorithm. (**C**) The position of the tip of the tail over time for the videos in (**A**). (**D**) The distribution of tail beat amplitudes for each bout in expert-classified prey capture and spontaneous swim videos. (**E**) Duration of the
*Figure 1. Continued on next page*

*Figure 1. Continued*

longest bend greater than 20° during each bout. (**F**) Overview of support vector machine (SVM) based bout classification procedure, displaying only two parameters (maximum tail bend and mean tail tip deflection) for clarity. Bouts are extracted using a threshold on the normalized and smoothed first derivative of tail bend angles. Values for each parameter are calculated for all bouts and used to train an SVM. The SVM is then used to classify unlabeled bouts. See *Figure 1—figure supplement 1* for plots of each of the five parameters, and accuracy of the SVM vs number of parameters.

The following figure supplement is available for figure 1:

**Figure supplement 1**. Prey capture and spontaneous behavior can be classified using five parameters.

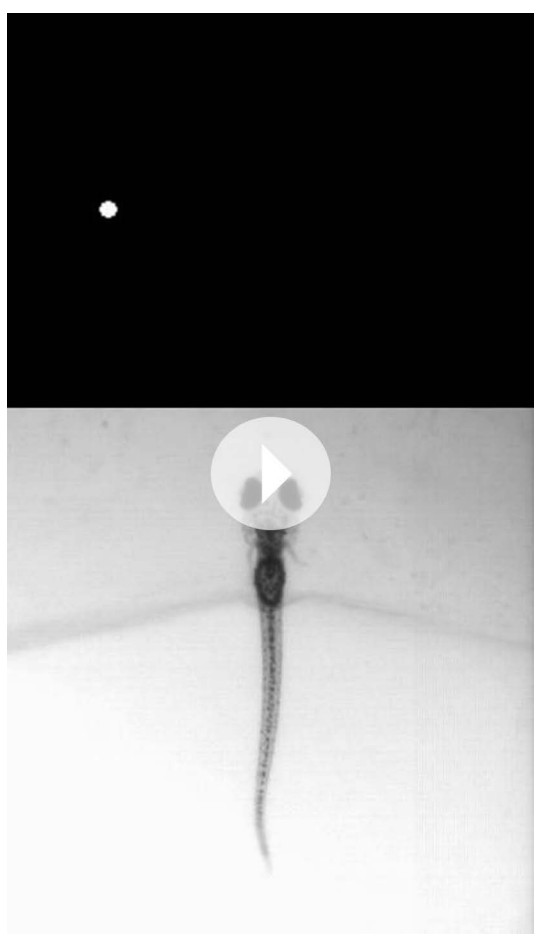

**Video 1**. Prey capture in response to virtual prey. A head-fixed larva responds to a 3° dot moving at 90°/s.

(*Gabriel et al., 2012*; *Nikolaou et al., 2012*), which makes it difficult to determine whether the tectal RGCs are responding selectively to prey. We thus focused our imaging experiments on AFs 1–9. We performed two-photon imaging of RGC axons in larvae with the calcium indicator GCaMP6 driven by an RGC-specific promoter. In these larvae, the ten AFs can be identified as distinct regions of fluorescent neuropil (*Figure 3A,C*). We asked whether the axons in any of these regions would respond to the ideal prey stimulus identified in our behavioral assay (3°, 90°/s). In our preparation, we could image all but the most ventral arborization field, AF1. We stimulated larvae with the prey stimulus while imaging RGC axons in AFs 2–9, and found one area, AF7, whose RGC axons responded robustly to the stimulus (*Figure 3B*), while seven other AFs did not respond to virtual prey. Some RGC axons in the optic tectum responded to the 3° stimulus, but the tectum response was much larger for stimuli of >10° (*Figure 3—figure supplement 1*). We next varied the size of the stimulus while imaging in the AF7 plane, and found that AF7 RGC axons responded maximally to dots 2–3° in diameter, and much less to stimuli larger than 6° in diameter (*Figure 3C–I*). Interestingly, AF9, another AF in the same plane, did not respond to small dots but began to respond as the stimulus size was increased above 6° in diameter (*Figure 3C–F*). We also measured the response of AF7 to stimuli of varying speeds, and found that the optimal speed was 90°/s, and stimuli slower than 12°/s or faster than 360°/s evoked

a much smaller response in these axons (*Figure 3G*). These tuning curves are strikingly similar to the behavioral tuning curves for size and speed (*Figure 2A,B*, grey trace in *Figure 3F,G*). At speeds above 360°/s the strong correlation between behavioral tuning and AF7 response degrades, since the behavioral response to extremely fast speeds is quite low. This could be caused by a threshold or non-linearity in the neural circuit, such as the weak activation of the AF7 RGC axons being insufficient to drive the downstream neurons. We next tested the direction selectivity of the axons in AF7, and found that the area as a whole responded to prey stimuli moving in all four cardinal directions (*Figure 3I*), although the response was more robust to stimuli moving horizontally (p < 0.001, *t*-test).

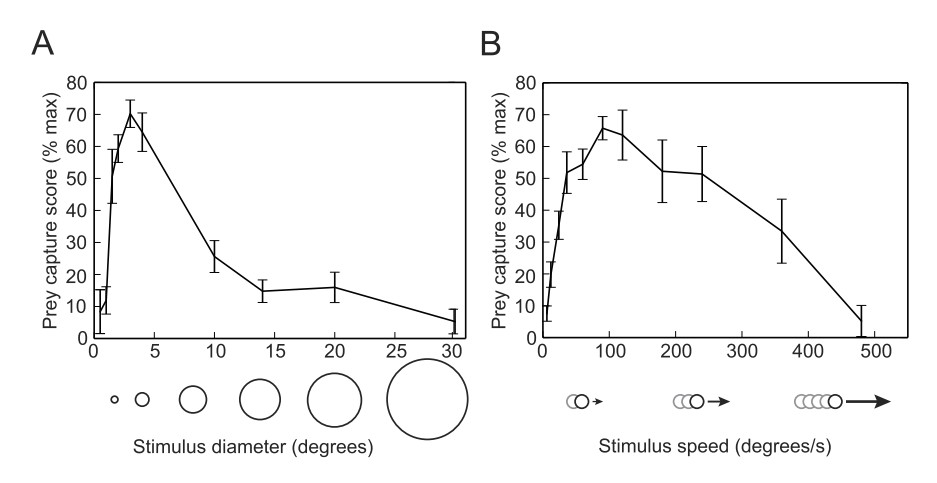

**Figure 2**. Prey capture behavior is triggered by dots of a particular size and speed. (**A**) Larvae were shown white dots of various sizes moving at 90°/s. Trials were scored by using the SVM to classify each bout and calculating the percentage of the trial that consisted of prey capture bouts. Scores are expressed as a percentage of the maximum for that larva. n = 16 larvae. (**B**) Prey capture behavior in response to a 3° dot moving at 6 to 480°/s. n = 9 larvae. Error bars = ±SEM.

## Response to paramecia

If AF7-projecting RGCs are mediating behavioral responses to virtual prey, we reasoned that they should also respond to natural prey, such as paramecia. We found that head-fixed larvae will perform j-turns and forward swims when paramecia are added to the surrounding medium (*Figure 4A*). In the absence of paramecia, head-fixed larvae perform spontaneous swims but rarely prey capture (*Figure 4B*), while after paramecia were added to the dish we observed frequent prey capture bouts (*Figure 4C*). Like the behavior evoked by virtual prey, these swims could be reliably classified by an SVM. We next asked whether the paramecia were activating AF7 RGC axons. We recorded baseline activity for 30 s and then added a drop of a dense paramecia culture to the petri dish. We found that AF7 RGC axons were strongly activated during the paramecium trials, whereas AF9 axons were not (*Figure 4D*, *Video 2*). When we quantified the ΔF/F in AF7 and AF9 for nine fish, we found that the peak response was significantly higher in AF7 during trials with paramecia (*Figure 4E*). The AF9 peak response in the same larvae was not significantly different in the presence of paramecia. The finding that AF7 axons respond to actual prey supports a role for this area in generating prey capture behavior.

## Anatomy of AF7-projecting RGCs

The activation of AF7 RGC axons specifically by small moving dots and paramecia suggests that there could be a specialized class of RGCs that project to this area. A subpopulation of RGCs has been previously shown to innervate AF7 and the *stratum opticum* (SO), the most superficial layer of the tectum (*Xiao and Baier, 2007*). However, little is known about their dendritic morphology, and it is unclear if these are the only inputs to AF7. We set out to characterize the anatomy of the AF7-targeting RGCs using BGUG (*Xiao and Baier, 2007*), a highly variegated *UAS:mGFP* reporter that allows the visualization of individual neurons within a Gal4 line. By expressing mCherry and BGUG in RGCs, we labeled most RGCs red and <1% of them green. We could thus identify single RGCs with arbors in AF7 (*Figure 5A*) and trace the axon back to the retina to observe the cell's dendritic morphology. We identified 19 larvae with an arbor in AF7. All 19 RGCs exhibited the same axonal morphology, projecting to AF7 and the SO, but not to any other AFs (*Figure 5A,B*). Thus, a prey-specific information channel is routed to two distinct visual areas by each axon. Examining the dendritic morphology of these AF7-targeting RGCs revealed that AF7 receives inputs from two distinct RGC types. Thirteen cells (out of 19) had bistratified arbors with branches in ON and OFF layers of the inner plexiform layer (IPL) (*Figure 5C*), and the other six formed diffuse arbors spanning the entire IPL (*Figure 5D*). These

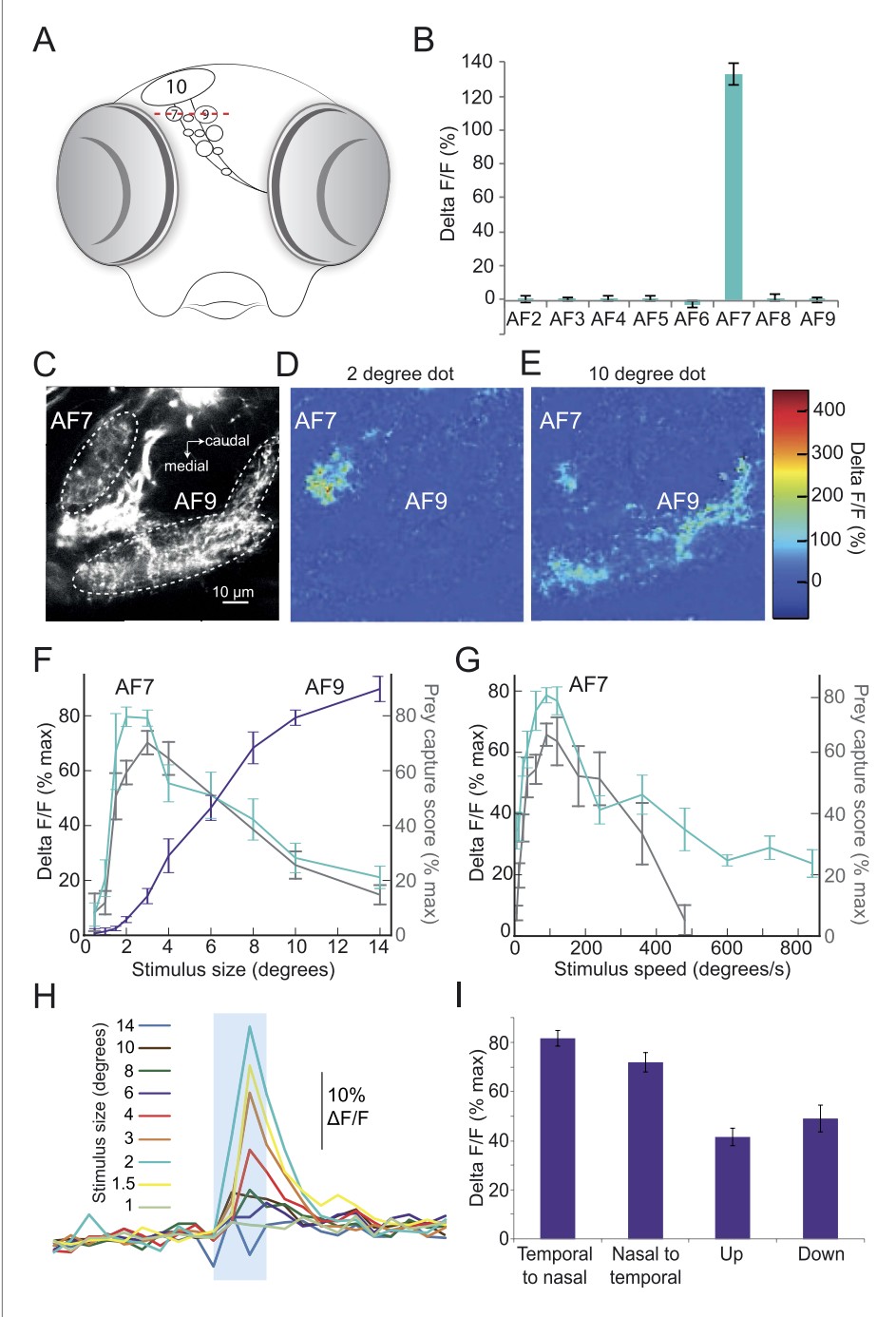

**Figure 3**. Prey stimuli activate RGC axons that project to AF7. (**A**) Schematic frontal view of the brain showing AFs. Red line indicates imaging plane for **C**–**G**. (**B**) Response to the ideal prey stimulus (3° dot, 90°/s) in other AFs in an *Isl2b:Gal4, UAS:GCaMP6s* transgenic larva. n = 9 larvae. (**C**) Baseline fluorescence of RGCs in an *Isl2b:Gal4, UAS:GCaMP3* larva. (**D**) Peak frame in the response to a 2° dot. (**E**) Peak frame in the response to a 10° dot. (**F**) Response of all larvae (n = 9) to stimuli 0.5–14° in diameter. ROIs were defined anatomically as in (**C**). The ΔF/F is plotted as a percentage of the maximum response for that larva. Grey lines represent the behavioral tuning curve from *Figure 2A*. (**G**) Response of AF7 RGC axons to a 3° dot travelling at a speed of 6–800°/s. Grey lines represent the behavioral tuning curve from *Figure 2B*. (**H**) Responses of AF7 to a range of different size stimuli. Blue box represents the one second interval when the stimulus was onscreen. (**I**) AF7 response to 3° dot moving in

*Figure 3. Continued on next page*

*Figure 3. Continued*

various directions. n = 10 larvae. Error bars = ±SEM. See *Figure 3—figure supplement 1* for tuning properties of tectal RGC axons.

The following figure supplement is available for figure 3:

**Figure supplement 1**. Response of RGC axons in the tectum to stimuli of varying sizes.

---

dendritic morphologies correspond to two of the 14 RGC classes previously described in the zebrafish retina (*Robles et al., 2014*), known as B2 and D1 These two types also project to other AFs, in much smaller numbers. However, the B2 and D1 neurons that project to AF7 only arborize in AF7 and the tectum. The RGC axon responses to prey stimuli in AF7 could be due to either B2 or D1 neurons, or both, as our Gal4 line labels both types.

## Ablation of AF7 neuropil

To test whether the AF7-projecting RGCs play a role in prey capture behavior, we performed two-photon laser ablations targeting the AF7 neuropil. Ablations were performed bilaterally in larvae expressing the fluorescent protein Dendra in RGCs (*Figure 6A*). Within 3 hr, degeneration of axons that project to AF7 could be observed (*Figure 6B*). Ablation of AF7 did not detectably change the projection of nearby RGC axons on their course to the tectum, nor did it decrease the intensity of RGC

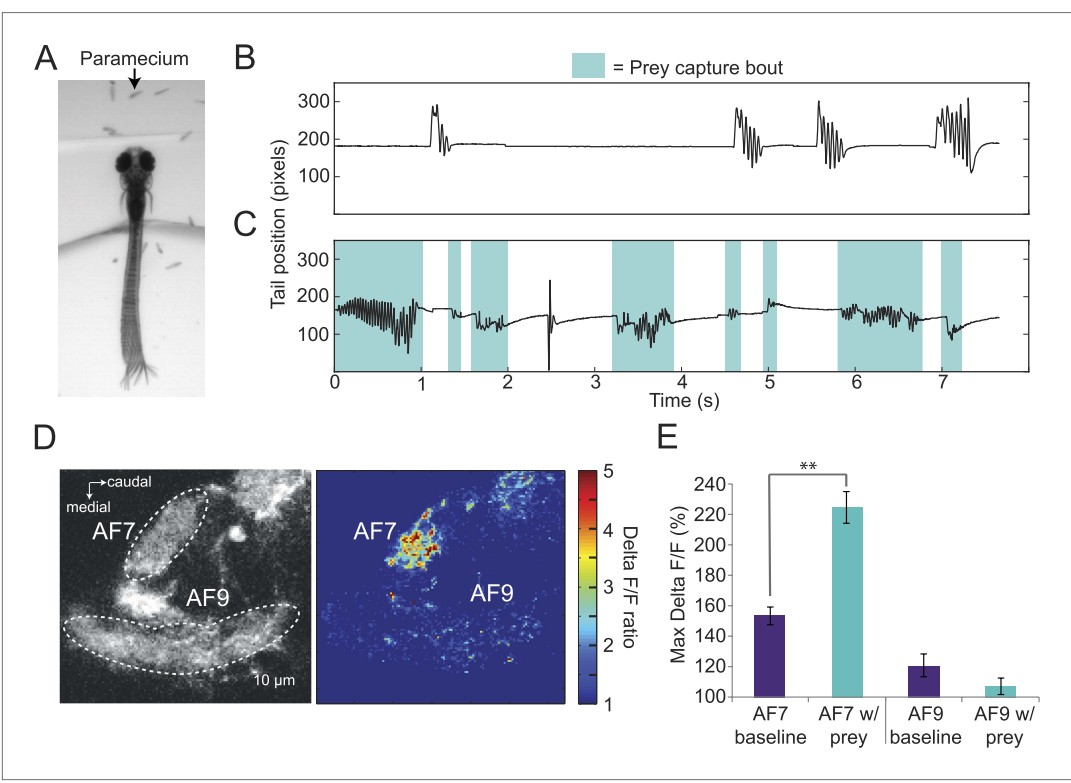

**Figure 4**. Paramecia evoke prey capture behavior and a response in AF7. (**A**) Overlay of 50 frames (167 ms) of high-speed video showing a head fixed larva responding to paramecia. (**B**) The position of the tip of the tail of the larva in A without paramecia, showing spontaneous swims. (**C**) The same larva after paramecia were added to the dish. Bouts that were identified as prey capture by the SVM are colored blue. (**D**) *Ath5:Gal4, UAS:GCaMP6s* larvae were imaged before and after addition of paramecia. Pseudocolor represents the ratio of the ΔF/F with paramecia to without. (**E**) AF7 and AF9 responses to paramecia in nine larvae. Maximum ΔF/F is plotted for each trial. The AF7 response was significantly higher in trials with paramecia (p = 9.6 × 10⁻⁵, Wilcoxon rank sum test), whereas the AF9 response was not significantly different (p = 0.083). n = 10 larvae. Error bars = ±SEM. See also *Video 2*.

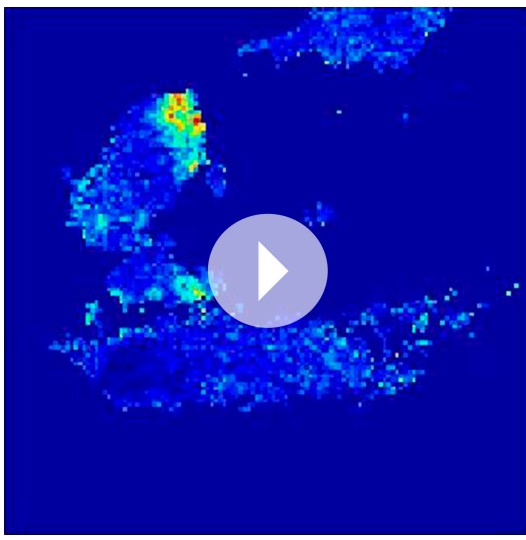

**Video 2**. AF7 RGC axons respond to paramecia. *Ath5:Gal4, UAS:GCaMP6s* axons are activated in the presence of paramecia. Pseudocolor represents the ratio of the ΔF/F with paramecia to without. Scale is the same as in *Figure 4D*. Imaging frame rate was 3.3 Hz.

axon labeling in the SO (*Figure 6—figure supplement 1A–D*). We also observed that tectal neuron responses to small stimuli remained intact after AF7 ablation (*Figure 6—figure supplement 1E–H*). These findings suggest that the ablation was restricted to the AF7 neuropil.

We recorded responses to swimming paramecia for 5 min before the ablation and again several hours afterwards, and used our SVM to categorize the bouts (*Figure 6C,D*). As a control, we ablated a similarly sized region of AF9. We found that larvae with AF9 ablations spent as much time performing prey capture bouts as before ablation, whereas in AF7-ablated larvae prey capture time was significantly reduced (*Figure 6E,F*). On average, the AF7-ablated larvae spent 40% as much time performing prey capture bouts as they did before the ablation (*Figure 6G*). Although the effect was robust, ablated larvae did perform some prey capture bouts, which could reflect a failure to ablate all AF7 RGC axons, or a contribution from other retinal inputs to the behavior. To confirm that the AF7 ablation did not generally impair visual function, we tested the optomotor response (OMR) before and after ablations in AF7 or AF9. We found that, consistent with previous work (*Roeser and Baier, 2003*) the OMR was not impaired in AF7-ablated larvae, but was reduced in AF9-ablated larvae (*Figure 6H*). These data suggest that AF7-projecting RGC axons are specifically required for prey capture behavior.

## Pretectal neurons with arbors in AF7 project to the tectum, nMLF and hindbrain

We used single cell electroporation to identify AF7's putative postsynaptic partners, by targeting cell bodies near AF7. We identified two classes of neurons with a proximal arbor located in the AF7 neuropil and distal arbors in other brain areas. The first class of neurons with processes in AF7, as well as the neighboring non-retinorecipient neuropil, formed a distinct projection to the tectum (*Figure 7A,B*). In all cases (n = 12) the tectal branch was located in a region between the SO and SFGS layers that does not receive RGC axons. We also observed that all of these projections terminated in the anterior fifth of the tectum (*Figure 7B,C*, and *Figure 7—figure supplement 1*).

A second type of neuron with dendrites in AF7 formed a projection to the nucleus of the medial longitudinal fasciculus (nMLF) and hindbrain (*Figure 7D–F*, and *Figure 7—figure supplement 1*). The line *Gal4^s1171t*, which labels neurons in the midbrain tegumentum, including the nMLF (*Thiele et al., 2014*), was used as a landmark to identify the nMLF. These neurons (n = 4) projected to the ipsi- and contralateral nMLF before terminating in the contralateral hindbrain. Thus, these AF7-innervating neurons could be directly involved in generating prey capture swims and j-turns.

## Discussion

While it was previously established that larvae respond to small moving stimuli (*Bianco et al., 2011*; *Trivedi and Bollmann, 2013*), little was known about where in the visual system this selectivity is generated. Our finding that a subset of RGC axons is tuned to prey stimuli suggests that the selectivity is generated early in the visual pathway, by circuits in the retina. There are examples of retinal circuits that generate a similar degree of stimulus selectivity. Experiments in the rabbit have identified local edge detectors—RGCs that respond only to slowly moving targets less than 1° in diameter (*Levick, 1967*), and a similar type has been found in the mouse retina (*Zhang et al., 2012*). The unusually strong surround suppression in these cells is thought to be a consequence of amacrine cells acting presynaptically to inhibit bipolar cells as well as RGCs (*van Wyk et al., 2006*; *Russell and Werblin, 2010*). A similar mechanism could account for the size tuning of AF7-projecting RGCs.

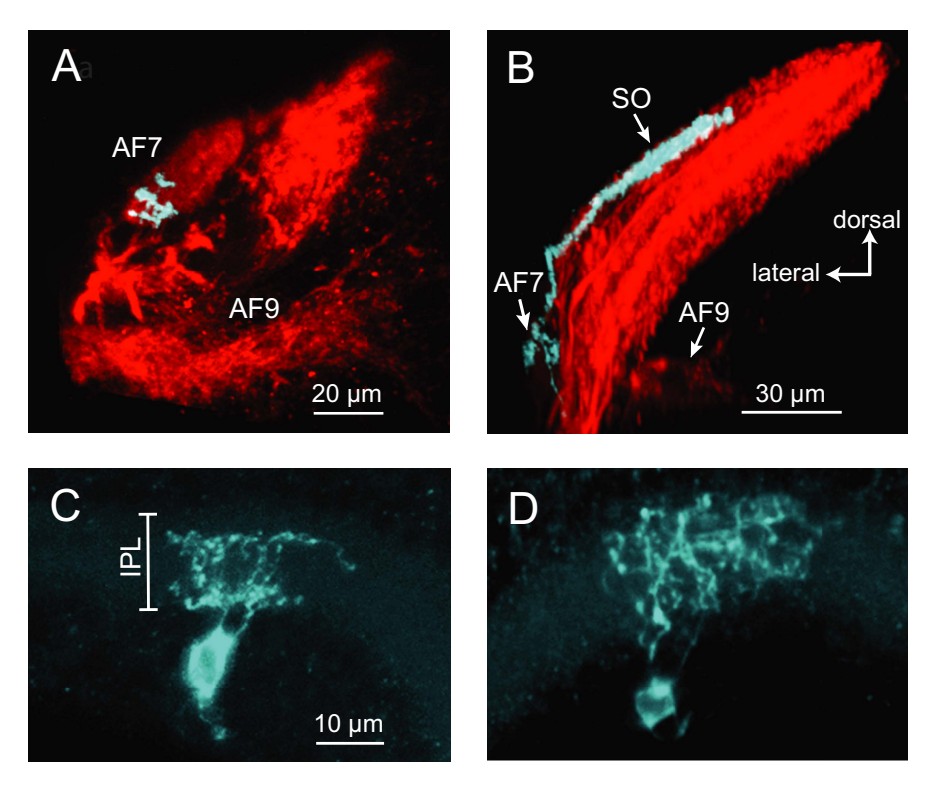

**Figure 5**. Two types of RGCs project to AF7 and also arborize in the tectum. (**A**) A single GFP-expressing RGC axon (cyan) arborizes in AF7 in an *Ath5:Gal4, UAS:mCherry, BGUG* larva. (**B**) The same axon also innervates the SO layer of the tectum. (**C**) Section of the retina showing dendritic morphology of a bistratified AF7-projecting RGC. Bracket indicates borders of the IPL (16 µM). (**D**) Dendritic morphology of a diffuse bistratified AF7-projecting RGC.

In many cases, RGCs detect features that have a clear survival value for the animal. For example, *fast-OFF* RGCs in the salamander anticipate the location of a moving object (*Berry et al., 1999*), a feature that could be used to predict the location of moving prey (*Leonardo and Meister, 2013*). A classic study in the frog found RGCs that responded to small (1–3°) objects moving though the visual field; these cells were hypothesized to act as 'bug detectors' (*Lettvin et al., 1959*). These examples show that the retina can perform computations to detect ecologically important objects, allowing RGCs to transmit pre-processed information to the brain to coordinate behavior. Our data support a model in which retinal circuitry confers prey selectivity on a subset of RGCs, which transmit this information to two visual areas in the midbrain: AF7 and the SO layer of the optic tectum.

Based on the axonal branching pattern of its RGCs, we can identify AF7 as the parvocellular superficial pretectal nucleus (PSp). The PSp is a retinorecipient brain area innervated by RGC axons that also form an arbor in the most superficial retinorecipient layer of the tectum in goldfish (*Springer and Mednick, 1985*). PSp was one of the two possible adult brain areas suggested for AF7 based on its location in classical dye tracing work (*Burrill and Easter, 1994*). This nucleus was originally identified as the homologue of the lateral geniculate nucleus (LGN) in the mammalian thalamus (*Schnitzlein, 1962*), although this is controversial (*Northcutt and Butler, 1976*). PSp has been anatomically described in several fish species, and is known to receive topographically organized retinal input (*Springer and Mednick, 1985*), although little functional information is available. Our group has recently demonstrated that AF7/PSp contains a retinotopic map. Most of the RGCs that project to this area reside in the temporal retina, creating a high-resolution representation of the anterior visual field (*Robles et al., 2014*). This is the part of the visual environment in which the prey is usually located during prey capture behavior (*Patterson et al., 2013*).

Why might it be advantageous to route prey-responsive RGCs to both AF7 and the tectum? One possibility is that tectal neurons are responsible for directing the orienting movements towards a particular point in visual space, while AF7 neurons modulate or gate the tectal output, based on

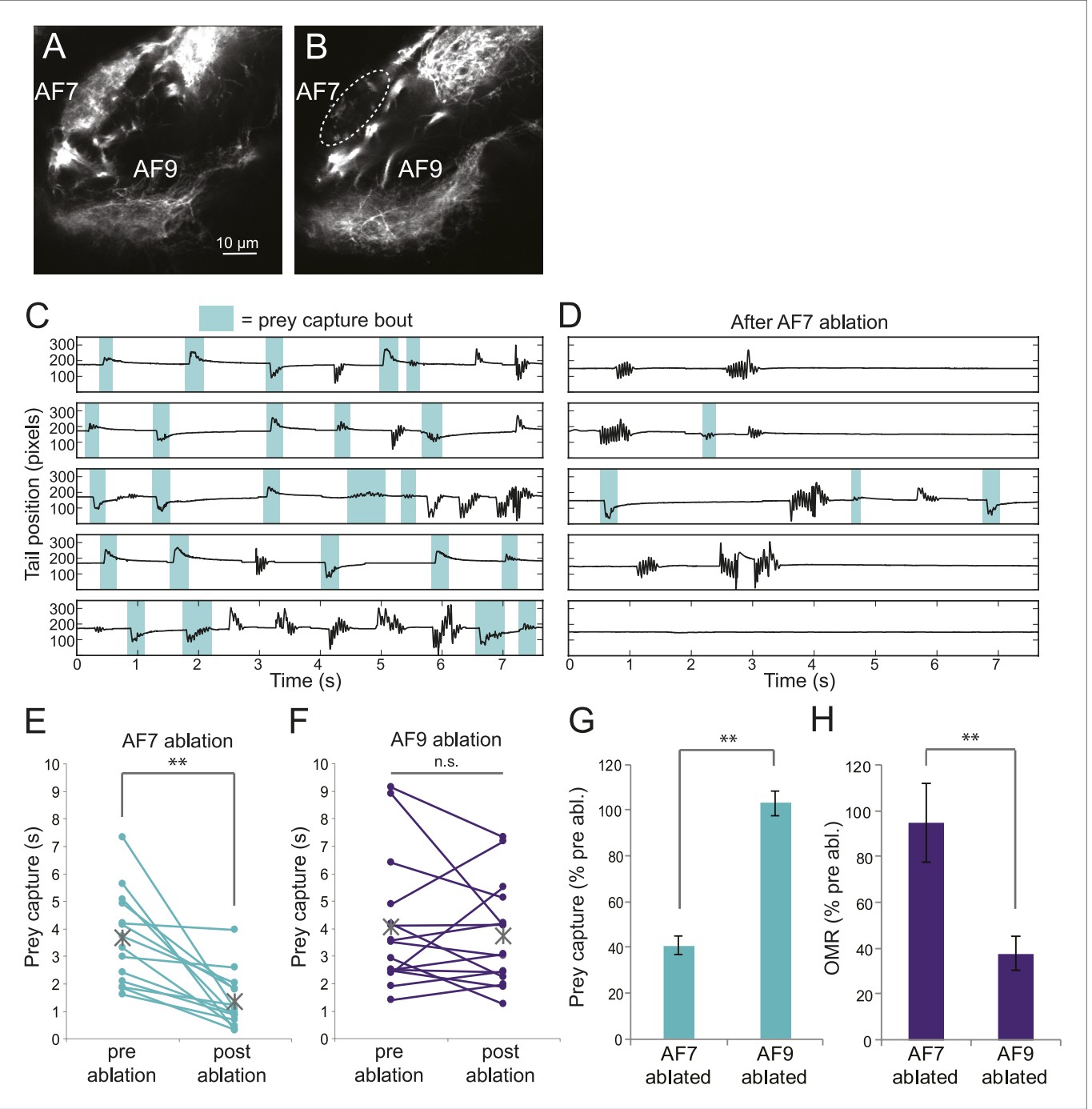

**Figure 6**. Ablation of AF7 markedly reduces prey capture behavior. (**A**) Intact *Ath5:Gal4, UAS:Dendra* larva at 8dpf. (**B**) *Ath5:Gal4, UAS:Dendra* larva after ablation of AF7 neuropil. (**C**) Pre-ablation response to paramecia in one example larva. Behavior was recorded for 5 min and the videos were edited to display all swim bouts. Each bout was classified by the same SVM as in *Figure 4C*. (**D**) The same larva after 2P laser ablation of AF7. (**E**) Total duration of prey capture bouts during the five 1 min trials before and after AF7 ablations. n = 14 larvae. p = 2.59 × 10⁻⁴, Wilcoxon rank sum test. (**F**) Prey capture time before and after AF9 ablations. n = 15 larvae. (**G**) Duration of prey capture bouts after ablation as a percent of initial prey capture time. p = 2.68 × 10⁻⁴, Wilcoxon rank sum test. (**H**) Optomotor response to moving gratings after AF7 and AF9 ablation. AF7ablation n = 7 larvae, AF9 ablation n = 9 larvae. p = 5.2 × 10⁻³, Wilcoxon rank sum test. Error bars = ±SEM. See *Figure 6—figure supplement 1* for AF7 and SO axon anatomy and periventricular neuron activity before and after ablation.

The following figure supplement is available for figure 6:

**Figure supplement 1**. RGC axons in AF7 and the tectum after AF7 ablation.

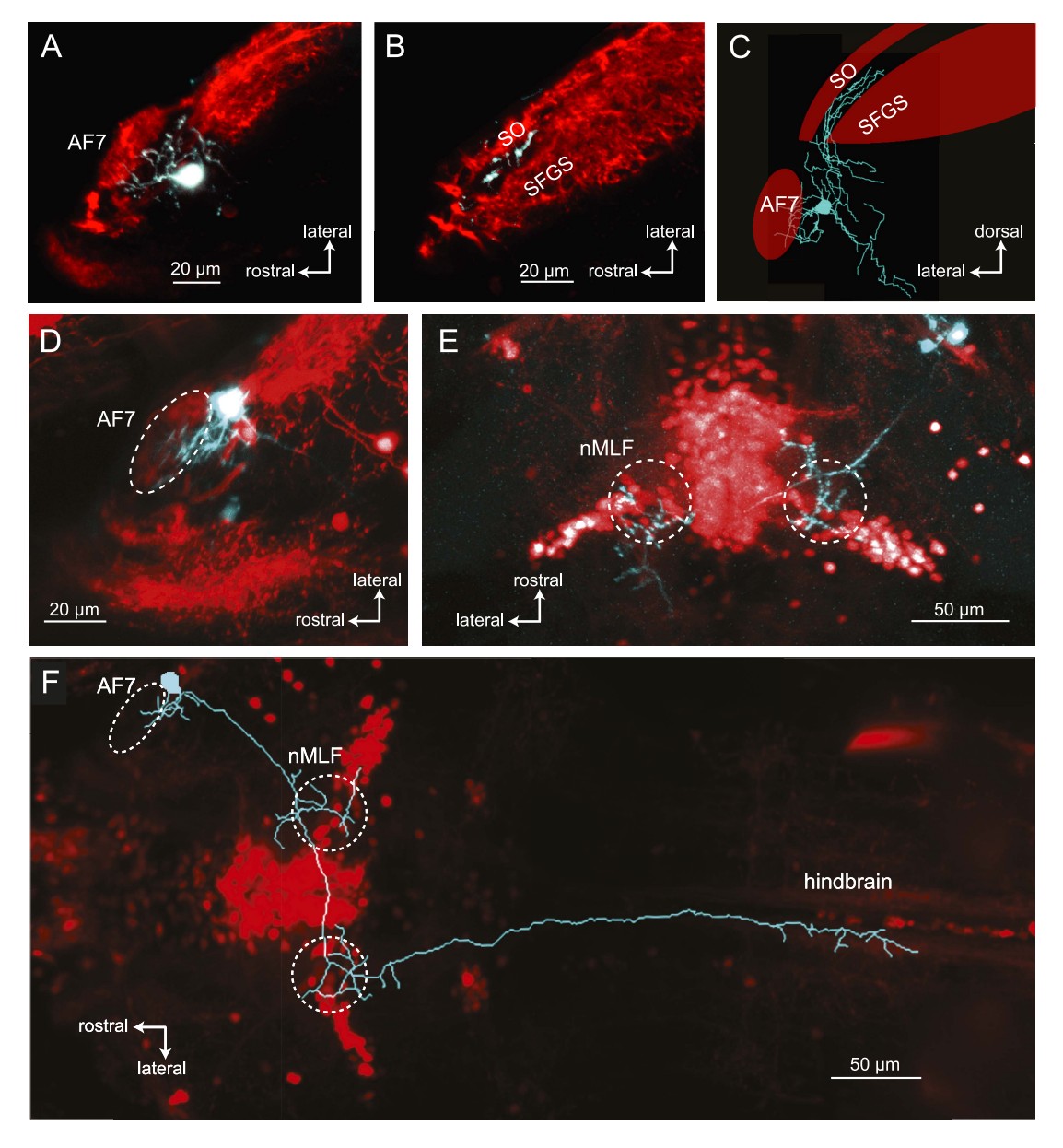

**Figure 7**. Morphologies of pretectal AF7 neurons. (**A**) Single cells were electroporated with tetramethylrhodamine (TMR) dextran (cyan) in an *Ath5:Gal4, UAS:GCaMP6s* larva (red). These neurons innervate AF7, as well as the adjacent non-retinorecipient neuropil in the same plane. (**B**) The cell imaged in (**A**) projects to the rostral tectum. (**C**) Tracing of the same cell, showing pretectal and tectal arbors. (**D–E**) Confocal images of a single electroporated cell in an *Ath5:Gal4, Gal4^s1171t UAS:GCaMP6s* larva. (**F**) Tracing of the same cell as in (**D**) and (**E**) overlaid on a confocal image showing the *Gal4^s1171t* labeling to identify the nMLF. See *Figure 7—figure supplement 1* for more examples.

The following figure supplement is available for figure 7:

**Figure supplement 1**. Examples of AF7 neurons.

behavioral state (e.g. hunger, arousal). The tectum and its mammalian homologue, the superior colliculus, have been shown to be involved in the transformation of sensory information encoded in terms of visual space into the appropriate motor command (*Bergeron et al., 2003*; *du Lac and Knudsen, 1990*). The tectum has also been implicated in zebrafish prey capture in several recent studies. Ablation experiments have shown that the tectal neuropil and, more specifically, inhibitory

tectal neurons in the most superficial layer play a role in prey capture (*Gahtan et al., 2005*; *Del Bene et al., 2010*), and functional imaging experiments have found that tectal neurons are activated by moving paramecia (*Muto et al., 2013*). Thus, the tectal circuits could translate a prey object's precise location in visual space into a swimming movement that results in pursuit of the prey.

We have identified a class of pretectal neurons that have dendrites in the AF7 neuropil and also project to the most rostral fifth of the tectum (*Figure 7A–C*, *Figure 7—figure supplement 1*) which corresponds to the anterior visual field. If the motor map is roughly faithful to the visual map, tectal output neurons in this region could trigger prey capture forward swims. Within the pretectum, the AF7/tectum neurons also have processes in the non-retinorecipient neuropil next to AF7, which most likely corresponds to the magnocellular superficial pretectal nucleus (PSm). We hypothesize that these neurons could be integrating the retinal response to prey with information on behavioral state, and then providing excitatory input to the tectum when appropriate. These AF7 neurons project to a layer of the tectum just ventral to SO, the tectal layer that receives prey-responsive RGC axons (*Figure 5B*, *Figure 8*). Tectal neurons with dendrites in these superficial layers could thus receive positional information from the raw retinal input in the SO, and excitatory input from the AF7 neurons that would release prey capture behavior only under certain conditions. However, at this point we do not know the response properties or type of neurotransmitter released by these neurons, so we can only speculate as to the input they provide to the tectum.

Modulation of the tectum by other visual areas is a common motif in vertebrate systems, and in many cases suppression or facilitation of tectal activity determines whether a behavioral response will occur. For example, a GABAergic projection from the basal ganglia to the superior colliculus is thought to act as a gate for the saccade-generating neurons of the superior colliculus, providing tonic inhibition whose cessation releases saccade movements (*Shires et al., 2010*). Another example of tectal modulation is the local cholinergic and global GABAergic feedback to the tectum provided by the isthmic nuclei in birds (*Knudsen, 2011*). The focal enhancement by cholinergic input has been shown to be necessary for the transmission of visual information to the area downstream of the tectum (*Marín et al., 2007*). Finally, studies of prey capture in toads have identified a pretectal area that provides inhibitory input to the tectum, resulting in the preference for worm-like prey stimuli (*Ewert et al., 2001*). The sign of this modulation is opposite to that described in our model, but this might be explained by the different types of stimuli being selected by the two systems (i.e., elongated objects moving in the direction of their long axis, vs small moving circles).

In addition to the class of AF7-innervating neurons that project to the tectum, we found another class which projects bilaterally to the nMLF and contralaterally to the caudal hindbrain (*Figure 7D–F* and *Figure 7—figure supplement 1*). The nMLF is a cluster of reticulospinal neurons that are involved in controlling swim

**Figure 8**. Model for prey capture circuitry. A prey stimulus on the left activates RGCs in the left eye (blue) which project to AF7 and the SO layer of the right tectum. Pretectal neurons (red) arborize in AF7 and send projections to the tectum or the nMLF and hindbrain. Activation of this circuitry produces a j-turn to the left, turning the larva in the direction of the prey. Previous studies have demonstrated connections between the tectum and nMLF (*Gahtan et al., 2005*) and between the tectum and hindbrain (*Robles et al., 2011*) (gray arrows).

orientation (*Thiele et al., 2014*), as well as speed (*Severi et al., 2014*). Two of the identified nMLF cells, MeLr and MeLc, have also been shown to be necessary for successful prey capture (*Gahtan et al., 2005*). Projections from AF7 to the nMLF could therefore be involved in generating prey capture tail movements. The hindbrain also plays an important role in locomotion. Neurons in the hindbrain project to the spinal cord, and activation of the caudal hindbrain has been shown to drive swimming behavior (*Arrenberg et al., 2009*; *Kimura et al., 2013*). Thus far, the tectum has been considered the main output pathway to premotor centers in zebrafish. However, in other systems, multiple areas project to the premotor nuclei and may drive behavior independently or direct different aspects of a behavior; for example, in primates, both the frontal eye fields and the superior colliculus provide input to the brainstem regions controlling eye movements (*Scudder et al., 2002*). It remains to be seen what role the neurons projecting from AF7 to the nMLF and hindbrain might play in the various components of prey capture behavior.

Our results identify a specific pretectal, retinorecipient region as a key component of the prey detection circuit. They also point toward a series of parallel and interconnected pathways between multiple areas involved in this elementary form of object recognition. Finally, they identify potential anatomical links of the visual network to midbrain and hindbrain areas that serve to coordinate the locomotor maneuvers involved in capturing prey. Further studies should shed light on what role each area plays in the behavior and how they interact.

## Materials and methods

### Fish

Embryos were raised in Danieau's solution (17 mM NaCl, 2 mM KCl, 0.12 mM MgSO$_4$, 1.8 mM Ca(NO$_3$)$_2$, 1.5 mM HEPES) at 27.5°C on a 14/10 light/dark cycle. Wild type TL larvae were used for behavioral experiments, and TLN larvae (*nacre* mutants) were used for imaging. Since gonadal differentiation has not occurred at this stage, males and females were used indiscriminately. All animal procedures conformed to the institutional guidelines of the Max Planck Society and the local government (Regierung von Oberbayern).

### Transgenic lines

The following transgenic lines were used: *Tg(Atoh7:Gal4-VP16)s1992t* (written as Ath5:Gal4), *Tg(UAS:Dendra-Kras)s1998t*, *Tg(UAS:mCherry)*, *Tg(pou4f3:Gal4, UAS:mGFP)* (a.k.a. BGUG), *Gal4$^{s1171t}$*, *Tg(Isl2b.2:Gal4-VP16)* and *Tg(elavl3:GCaMP5G)*. To construct the *UAS:GCaMP6s* line, a fragment encoding GCaMP6s (Addgene) was cloned into a *14xUAS, pTol2* plasmid.

### Behavioral experiments

At 5 dpf, each dish of larvae was fed about 2 ml of a dense culture of paramecia. At 6 dpf, the larvae were embedded in 2.5% low melting point agarose (Invitrogen, Carlsbad, California) in Danieau's solution and positioned 3 mm from the edge of a 35 mm Petri dish lid. After the agarose had set, the dish was filled with Danieau's solution and some of the agarose was cut away, leaving the tail caudal to the swim bladder free to move. After embedding, larvae were kept at 27.5°C for 48 hr. Behavioral experiments were conducted at 8 dpf in a 27.5°C chamber. For virtual prey experiments, larvae with a prey capture score (percent of the trial spent in prey capture bouts) of less than 30% for the 3° stimulus were excluded from the analysis.

Visual stimuli were designed using a custom program in the Python-based Vision Egg software (*Straw, 2008*). The stimulus was a white dot on a black background, moving horizontally from left to right. Stimuli were presented on a 12 × 9 mm OLED Microdisplay (Emagin, Bellevue, WA) covered with three magenta Wratten filters (Edmund Optics, Barrington, NJ) and positioned 1 cm from the larva. Larvae were illuminated from below with an IR light source. We used a high-speed camera (Photonfocus, Switzerland) to record tail movements at 300 frames/second (f/s), with a resolution of 300 × 300 pixels.

For behavioral experiments with paramecia, we embedded 6 dpf larvae in the center of a 35 mm Petri dish and cut away some of the agarose in front of the eyes by making an incision perpendicular to the body axis 1 mm from the head. The tail below the swim bladder was also freed. As we did for the virtual prey experiments, we waited 48 hr after embedding to test behavior. We added a few drops of paramecia (*Paramecium multimicronucleatum*, Carolina Biological Supply Company, Burlington, NC) to the dish and recorded at 300 f/s for 5 min. The procedure was repeated a few hours after laser

ablations. For post-ablation behavioral experiments, the experimenter was blinded as to whether the animal was in the AF7 or AF9 ablation group.

To test the OMR, each embedded larva was positioned in an arena surrounded by three LED screens (5.5 × 7.5 cm). Gratings moving leftward at 20°/s were displayed on all three screens to produce turns. Stimulus presentation was controlled with a custom LabVIEW script, and tail movements were recorded at 250 f/s. The same procedure was repeated a few hours after laser ablations. To measure the number of swims, we digitized the tail as described below and used a thresholding operation on the smoothed derivative of the tail bend angle to identify swim bouts.

## Tail digitization

We developed a custom Python program that quantifies the tail position in each frame as a series of points, with approximately 40 points covering the tail. The program uses the OpenCV library to load videos. At the start of each video, the user is queried to select the base of the tail. From that point, the program iterates through the tail in each frame, taking a slice through the image where the next tail midpoint is expected to be. This slice is smoothed, convolved with a tail-like kernel, and the maximum is taken to determine the midpoint of the tail. This sequence is repeated until the end of the tail, which is detected using a threshold for change in luminance across the slice. To improve accuracy, the width and contrast of the tail in the first frame of video are characterized and used to refine the tracking for the rest of video.

## Support vector machine (SVM)

The SVM code was written in Python. The SVM and cross-validation procedures were provided by the scikit-learn library (*Pedregosa et al., 2011*), which uses LIBSVM (*Chang and Lin, 2011*) for the SVM implementation. To avoid testing and training on the same dataset, we used fivefold stratified cross-validation for training (*Kohavi, 1995*). For this validation, data are split into five groups of approximately equal mean. One group is reserved for testing while the others are used for training, and this procedure is repeated five times before the results are combined. We quantified 16 parameters for each bout. A radial basis function (RBF) kernel shape was used to draw the decision boundaries. A subset of the 16 parameters was selected by plotting the accuracy vs number of parameters. The point that maximized performance while minimizing the number of parameters was selected. For the virtual prey SVM, five parameters were optimal (*Figure 1—figure supplement 1*).

The virtual prey SVM was trained on 248 prey bouts and 121 spontaneous bouts. The paramecia SVM was trained on 273 prey bouts and 396 spontaneous bouts.

For the parameters described below, 'tail angle' is the angle of deflection of tail center of mass, 'tail tip' is defined as the last eight points of tail, and 'tip angle' is the angle of a line dawn through the last eight points of the tail, with respect to vertical. Parameters 1–5 were used in the virtual prey SVM; parameters 1–6 were used in the paramecia SVM.

1. 'Maximum tail curvature': maximum over the bout.
2. 'Number of peaks in tail angle'.
3. 'Mean tip angle': absolute value of tip angle in each frame, averaged across the bout.
4. 'Maximum tail angle': maximum of the bout.
5. 'Mean tip position': average position of last eight points in the tail (horizontal deflection as a fraction of the tail length).
6. 'Number of frames between peaks': mean (over bout) number of frames between peaks in the tail angle.
7. 'Medium frequency power of tail angles': power of middle frequency band of fourier transform of tail angle.
8. 'Low frequency power of tail angles': power of low frequency band of fourier transform of tail angle.
9. 'Tail angle vs tip angle': mean (over bout) difference in angle of deflection between tail center of mass and tip center of mass.
10. 'Tail angle vs. tip angle at frame of maximum tail angle': difference in angle of deflection between tail center of mass and tip center of mass in the frame with the maximum tail angle.
11. 'Variance in tail angle': over entire bout.
12. 'Mean tail curvature': mean curvature over entire bout.
13. 'Mean tail tip angle': mean over entire bout
14. 'Mean tail position': average position of tail from $12^{th}$ point to end (horizontal deflection as a fraction of the tail).

15. 'Mean tail curvature vs. tail angle': mean (over bout) difference between tail curvature and tail angle.
16. 'Maximum tip horizontal deviation'.

## Two-photon calcium imaging

Larvae were embedded as for behavioral experiments. Imaging experiments were performed at 7 or 8 dpf. Stimuli were presented on the same type of OLED screen as for behavior, also covered with three magenta wratten filters. Imaging was performed on a movable objective microscope (Sutter, Novato, CA) using a 40× objective (Olympus, Japan). Excitation light was generated by a Ti: Sapphire laser (Coherent, Santa Clara, CA) tuned to 920 nm. Time series were recorded at 3.6 Hz with a resolution of 128 × 128 pixels. We used a red LED to provide some ambient light to the larva during imaging, and we heated the microscope box to 27°C.

## Confocal imaging

In order to record the response to paramecia in RGC axons, we embedded larvae in the center of a dish as for the paramecium behavior experiments and again cut away the agarose in front of the eyes and below the swim bladder. Since bright ambient light was required to allow the larvae to see the paramecia, we performed these imaging experiments on a confocal microscope (Zeiss 780) rather than a two-photon. Time series were acquired at a rate of 3.3 Hz and a resolution of 128 × 128 pixels. Experiments were performed with overhead lights on and a white LED illuminating the dish to enhance the visibility of the prey. Temporal series of the same field of view with and without paramecia were joined together. The resulting stack was registered with StackReg in Fiji to reduce motion artifacts and pixel misalignment between the two acquisitions. From the raw data, ΔF/F was computed pixel-wise by a custom Matlab routine. Because we did not know when a paramecium was within the field of view, we calculated the baseline F for each pixel as the value of that pixel during a period of low activity. This was defined as the eighth percentile of the pixel's fluorescence over a sliding window of 25 frames. Maximum intensity projections of ΔF/F, computed independently for the two segments of the stack (before and after addition of paramecia), were divided to give the ΔF/F ratio between the two conditions for the pseudocolor image shown in *Figure 4D*. From the ΔF/F maximum intensity projections, the average values over regions of interest corresponding to AF7 and AF9 were computed to compare the activity between the two arborization fields. During the 30 s windows we imaged over, occasional responses were observed in AF7 and AF9 in the absence of paramecia added to the bath. This could be the result of spontaneous activity, or small objects, for example bubbles or debris, floating in the solution.

## RGC anatomy

*UAS:mCherry, BGUG, Islet2b:Gal4* (or *Ath5:Gal4*) larvae were treated with 0.2 mM 1-phenyl-2-thiocarbamide (PTU) from 1 dpf to reduce pigmentation of the retina. Larvae with sparse GFP in AF7 were identified, and confocal stacks of the axonal projections were acquired. To visualize dendritic arbors in the retina, vibratome sectioning was performed as previously described (*Robles et al., 2013*). Briefly, 50 µm vibratome sections were cut from larvae embedded in gelatin/albumin/sucrose and stained with a chick anti-GFP primary antibody (Catalog number GTX13970; GeneTex, Irvine, CA) and an Alexa 488 anti-chick secondary antibody (Catalog number A-11039; Invitrogen).

## Two-photon laser ablation

Ablations were performed in 8 dpf *Ath5:Gal4 UAS:Dendra*, TLN larvae, using the same two-photon microscope as for the imaging experiments. We found that treating the larvae with 0.1 mM PTU starting at 1 dpf was helpful in preventing tissue damage during ablation. Axons were killed by scanning an 850 nm beam for 750 ms over a 3 × 3 µM area. Post-objective laser power was 80 mW/mm$^2$. 4–5 scans were usually sufficient to ablate the AF7 neuropil. A Z stack of the area was taken 24 hr later to assess completeness of the ablation, and larvae with less than 80% of the volume ablated bilaterally were not used for the analysis. As a control, we ablated a similar volume of AF9 neuropil, with 4–5 scans.

## Electroporations

6 or 7 dpf larvae were embedded in agarose, immersed in extracellular solution and anesthetized with 0.02% tricane. Larvae with the transgenes *Ath5:Gal4, UAS:GCaMP6s,* and for some experiments *Gal4$^{s1171t}$* were used. Patch pipettes (25–32 MΩ) were filled with 15% tetramethylrhodamine dextran

(3000 MW) in intracellular saline. AF7 and AF9 were used as landmarks to locate AF7 cells. Following contact with a cell, a voltage train was applied (1 s duration, 150 Hz, 1.4 ms pulse width, 2–6 V) by an Axon Axoporator (Molecular Devices, Sunnyvale, CA). Labeled cells were imaged on a Zeiss LSM780 confocal microscope.

## Statistical analysis

The Jarque–Bera test was used to determine whether data were normally distributed. The *t*-test was used to evaluate normally distributed data. For non-normally distributed data, we used the Wilcoxon rank sum test. Variance was tested and found to be similar between groups by Levene's test.

## Code Repository

All custom software is available at https://bitbucket.org/mpinbaierlab/semmelhack-et-al.-2014.

## Acknowledgements

We thank Estuardo Robles for help tracing neurons; Fumi Kubo for assistance with two-photon ablations; Marco Dal Maschio for help analyzing confocal data; and Cris Niell and members of the Baier lab for feedback on the manuscript.

## Additional information

### Funding

| Funder | Grant reference number | Author |
|---|---|---|
| Max-Planck-Gesellschaft | | Herwig Baier |
| National Eye Institute | EY12406 & EY13855 | Herwig Baier |
| Helen Hay Whitney Foundation | | Julia L Semmelhack |

The funders had no role in study design, data collection and interpretation, or the decision to submit the work for publication.

### Author contributions

JLS, Conception and design, Acquisition of data, Analysis and interpretation of data, Drafting or revising the article; JCD, Wrote custom software to analyze behavioral data, Analysis and interpretation of data, Drafting or revising the article; TRT, HB, Conception and design, Drafting or revising the article; EK, EL, Acquisition of data, Drafting or revising the article

### Ethics

Animal experimentation: All animal procedures conformed to the institutional guidelines of the Max Planck Society and the local government (Regierung von Oberbayern). The protocol (55.2-1-54-2532-101-12) was approved by the Regierung Oberbayern.

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
