## [Decision Letter]

Thank you for sending your work entitled “A dedicated sensorimotor pathway for prey detection and capture in larval zebrafish” for consideration at *eLife*. Your article has been favorably evaluated by Eve Marder (Senior editor) and 3 reviewers, one of whom is a member of our Board of Reviewing Editors.

The Reviewing editor and the other reviewers discussed their comments before we reached this decision, and the Reviewing editor has assembled the following comments to help you prepare a revised submission.

Semmelhack and colleagues, in their study in embryonic zebrafish establish that there is a retinal to pretectal connection that seems to convey sensory information from a small subset of retinal ganglion cells about prey items. Elimination of ganglion cell input to this pretectal area, AF7, strongly reduces prey capture behavior. These observations point to an initial sensory pathway that is focused on stimuli that represent prey and convey visual signals to AF7. The authors also attempt to anatomically link the sensory AF7 pathway with brainstem motor command systems. Altogether the authors suggest that they have identified a dedicated sensorimotor pathway for prey detection and capture in larval zebrafish.

In general the work is interesting. The strong and convincing part of the study is the characterizations of the initial sensory pathway in the prey capture behavior. The importance of study is that it appears to be the first time that feature selective ganglion cells activity has been linked to prey capture behavior outside the tectum. The less convincing part of the study is the link to motor output pathways. Moreover, the results appear to be oversold in places with too strong conclusions based on the available data. It is recommended that the authors' focus on what can clearly be said based on strong evidence and tone down strong statements and discussions based on weak evidence. The details are outlined below in addition to other points for improvements of the manuscript.

1) The title is: “A dedicated sensorimotor pathway for prey detection and capture in larval zebrafish.” The title is a statement that goes beyond the evidence. The sensory side seems fairly dedicated (although role of tectum needs to be considered), but the evidence for the motor end is weak.

2) It cannot be concluded, as in the Introduction—“(…) targeted laser ablation showed that this area is necessary for the behavior”— and otherwise, that AF7 is necessary for the behavior. As shown in the Results section, deletion of input to AF7 only reduces the prey capture behavior. Hence, it is not necessary. If the animals can do the prey capture behavior at all without the region then it is cannot be said to be “required”. A balanced discussion for the lack of necessity is required.

3) Please comment about the mismatch between behavioral response and AF7 at high speeds shown in Figure 3.

4) Please clarify if AF7 is only activated in prey capture behavior. For example, are escape responses observed and if so how do they relate to the parameters presented in Figure 2? Moreover, it seems that there are bigger AF7 responses when paramecia are present, but still there are responses even without paramecia. This would seem to refute the notion that the area is dedicated to prey responses, which is a basic conclusion of the work. If the area is responding when no prey stimuli are there, then the paper should make that clear.

5) How is the tectum responding to the optimal stimulus? (Please show traces and selectivity, as in Figure 3).

6) There are some ambiguities whether the two ganglion cells that are said to project exclusively to AF7 only project there (Figure 5) or if they also project to other AFs. Based on the data in Figure 5 in the authors Current Biology paper (2014), it appears that the two ganglion cell types that project to AF7 also project to other AFs. Please clarify this point.

7) The link between an anatomical class of ganglion cells (Figure 5) and the prey capture behavior is indirect. In a strict sense the authors have only shown that some ganglion cells activate the AF7 but not the type shown in Figure 5. This should be clear in the text.

8) The evidence for output pathways from the pretectal area consists of fills of a few neurons in the vicinity of the pretectal arborization region. It is not known whether the neurons are driven by the stimulus or whether they actually get inputs from the retina or have connections in their projection area. All of the output part, including several paragraphs of the Discussion section, are based on these findings. This is too weak a foundation for the speculations that a dedicated motor output pathway has been identified. Please adjust the text accordingly.

9) It is too strong to say “pinpoint the anatomical links of the visual network…”, since not all pieces have been found.

10) The statement “We identified a class of pretectal neurons that could function as a gate for tectal output” is highly speculative. This strong statement cannot be accepted as long as the authors do not provide any indication about the molecular identity of the found AF7-OT connection and the following discussion can be left out.

Other points for improvements of the manuscript:

1) Ewert's classic work addressed the innate capacity to respond to prey although in frogs/toads. Please cite his work in the initial opening sentences.

2) Figure 3 would be stronger by showing some real traces of responses from AF7.

3) Please comment about the mismatch between behavioral response and AF7 at high speeds shown in Figure 3.

4) What is the average size and the swimming speed of paramecia? It would be nice seeing it compared to the optimal behavioral parameters found.

5) Some more information should be provided about BGUG in the Results section in order to make the anatomy part more easily understandable for the lay reader.

6) The enlarged LGN in mammals can depend on a number of different causes and it might not be justified to stress the possible homology with the mammalian LGN in the Discussion.

7) From the sentences “From the raw data, ΔF/F was computed pixel-wise by a custom Matlab routine with F resulting from the eighth percentile of the sample distribution over a sliding window of 25 frames. Maximum intensity projections of ΔF/F, computed independently for the two segments of the stack, were divided to give the ΔF/F ratio between the two conditions” (Materials and methods section), it was hard to understand how exactly the data analysis was done. This needs to be more clearly explained. What is meant by sliding window here?

---

## [Author Response]

*1) The title is: “A dedicated sensorimotor pathway for prey detection and capture in larval zebrafish.” The title is a statement that goes beyond the evidence. The sensory side seems fairly dedicated (although role of tectum needs to be considered), but the evidence for the motor end is weak*.

We have changed the title to “A dedicated visual channel for prey detection in larval zebrafish”.

*2) It cannot be concluded, as done in the Introduction—“(…) targeted laser ablation showed that this area is necessary for the behavior”— and otherwise, that AF7 is necessary for the behavior. As shown in the Results section, deletion of input to AF7 only reduces the prey capture behavior. Hence, it is not necessary. If the animals* can *do the prey capture behavior at all without the region then it is cannot be said to be “required”. A balanced discussion for the lack of necessity is required*.

We have removed the two mentions of AF7 as necessary for the behavior, and added text to the Results section to discuss the presence of some prey capture in ablated larvae.

*3) Please comment about the mismatch between behavioral response and AF7 at high speeds shown in*
Figure 3*.*

We have added text to the Results section acknowledging and explaining the imperfect match.

*4) Please clarify if AF7 is only activated in prey capture behavior. For example, are escape responses observed and if so how do they relate to the parameters presented in*
Figure 2*?*

Escapes are triggered by expanding circles larger than 20° (our unpublished observation), which is similar to escape thresholds in other species. Our functional imaging data show that stimuli larger than 10° do not activate AF7 (Figure 3), which strongly suggests that AF7 is not involved in escape.

*Moreover, it seems that there are bigger AF7 responses when paramecia are present, but still there are responses even without paramecia. This would seem to refute the notion that the area is dedicated to prey responses, which is a basic conclusion of the work. If the area is responding when no prey stimuli are there, then the paper should make that clear*.

For the paramecium imaging experiment, the position of the free-swimming paramecia cannot be controlled. Therefore, we imaged over a relatively long time window (30 s), which ensured that paramecia would appear in the field of view. We then compared the time window with paramecia present to that without. When imaging with GCaMP6s over a long window, spontaneous responses are observed in many brain areas. It is also possible that small items floating in the solution (bubbles, debris) could occasionally activate AF7. We have added text to the Methods section to explain the presence of this activity.

*5) How is the tectum responding to the optimal stimulus? (Please show traces and selectivity, as in*
Figure 3*)*.

We have added a supplementary figure (Figure 3—figure supplement 1) showing tectal RGC axon size selectivity.

*6) There are some ambiguities whether the two ganglion cells that are said to project exclusively to AF7 only project there (*Figure 5*) or if they also project to other AFs. Based on the data in*
Figure 5
*in the authors Current Biology paper (2014), it appears that the two ganglion cell types that project to AF7 also project to other AFs. Please clarify this point*.

The two types of RGCs (B2 and D1, as classified by dendritic morphology) that project to AF7 also project to other AFs. However, the axonal type that projects to AF7 only innervates AF7 and the SO layer of the tectum. We now clarify these points in the Results section.

*7) The link between an anatomical class of ganglion cells (*Figure 5*) and the prey capture behavior is indirect. In a strict sense the authors have only shown that some ganglion cells activate the AF7 but not the type shown in*
Figure 5*. This should be clear in the text*.

It is true that we cannot distinguish whether one or both RGC types projecting to AF7, and shown in Figure 5, are responding to prey, since our Gal4 line labels both types of AF7 RGCs. We have clarified this in the Results.

*8) The evidence for output pathways from the pretectal area consists of fills of a few neurons in the vicinity of the pretectal arborization region. It is not known whether the neurons are driven by the stimulus or whether they actually get inputs from the retina or have connections in their projection area. All of the output part, including several paragraphs of the Discussion section, are based on these findings. This is too weak a foundation for the speculations that a dedicated motor output pathway has been identified. Please adjust the text accordingly*.

The filled neurons we show do have projections to the AF7 neuropil and overlap with the RGC axons of AF7 (e.g. Figure 7 A, C and D). We currently cannot determine if these neurons are indeed driven by the prey stimulus. We have therefore qualified our statements concerning the dedicated nature of the motor pathway that originates in AF7. We have also removed some of the discussion of the role of these neurons.

*9) It is too strong to say “pinpoint the anatomical links of the visual network…”, since not all pieces have been found*.

We have toned down this statement.

*10) The statement “We identified a class of pretectal neurons that could function as a gate for tectal output” is highly speculative. This strong statement cannot be accepted as long as the authors do not provide any indication about the molecular identity of the found AF7-OT connection and the following discussion* can *be left out*.

We have deleted that statement and modified this section of the Discussion to make clear that we do not know the molecular identity of the neurons and thus the sign of the modulation. We have also removed some of the speculation.

Other points for improvements of the manuscript:

*1) Ewert's classic work addressed the innate capacity to respond to prey although in frogs/toads. Please cite his work in the initial opening sentences*.

We now cite Ewert’s work at the beginning of the Introduction.

*2)*
Figure 3
*would be stronger by showing some real traces of responses from AF7*.

We have added a new panel to Figure 3 showing traces from one example larva (Figure 3).

*3) Please comment about the mismatch between behavioral response and AF7 at high speeds shown in*
Figure 3.

We have added text to the Results, acknowledging the imperfect match.

*4) What is the average size and the swimming speed of paramecia? It would be nice seeing it compared to the optimal behavioral parameters found*.

An accurate calculation of the size and speed of paramecia relative to the larva during prey capture would require careful observation and tracking of both free-swimming larvae and paramecia, and we would argue that this is beyond the scope of the paper. A recent paper from David McLean’s lab described the 2D location of larvae and prey before prey capture bouts, and found that paramecia were roughly 1-3 mm away when detected. Assuming that paramecia are about 150 µm long, that range would be 2-8° of the visual field along the paramecium’s long axis, which fits fairly well with our data.

*5) Some more information should be provided about BGUG in the Results section in order to make the anatomy part more easily understandable for the lay reader*.

We have added information about BGUG.

*6) The enlarged LGN in mammals* can *depend on a number of different causes and it might not be justified to stress the possible homology with the mammalian LGN in the Discussion*.

This sentence has been deleted.

7) From the sentences “From the raw data, ΔF/F was computed pixel-wise by a custom Matlab routine with F resulting from the eighth percentile of the sample distribution over a sliding window of 25 frames. Maximum intensity projections of ΔF/F, computed independently for the two segments of the stack, were divided to give the ΔF/F ratio between the two conditions” (Materials and methods section), it was hard to understand how exactly the data analysis was done. This needs to be more clearly explained. What is meant by sliding window here?

We have added text to the Methods section (under “RGC Anatomy”) to explain how we did the analysis.